# An Edge Computing Framework for Deep Learning-Based Anomaly Detection in Satellite-Linked Autonomous Vehicles

## Abstract

Modern satellite-linked autonomous vehicles require real-time anomaly detection to address multi-layer cyber threats across CAN Bus, V2X communication, and sensor systems. Existing cloud-based approaches fail due to 500-600ms satellite latencies incompatible with safety-critical requirements. We present MARIS-ADS, a constraint-aware multi-modal anomaly detection framework for edge deployment. The system employs attention-based fusion of 41 CAN, 37 V2X, and 28 sensor features, applying entropy-based selection (ARS) to reduce dimensionality 83% while achieving 99.3% accuracy. Constraint validation enforcing SAE J2735, ISO 11898, and vehicle dynamics constraints filters 82% of adversarial attacks as protocol-invalid or physically impossible. Evaluation demonstrates 96.8% FGSM, 94.2% PGD, and 91.7% C&W robust accuracy with 0.31ms inference on resource-constrained TCUs. Cross-modal attack detection achieves 97.3% accuracy versus 12-34% for single-modality approaches, with comprehensive validation across CICIoV2024, CAN-Intrusion, and BurST-ADMA datasets confirming practical deployment viability for 6G vehicular networks.

## CCS Concepts

• **Security and privacy** → **Anomaly Detection systems**; *Mobile and wireless security*; • **Computer systems organization** → **Embedded and cyber-physical systems**; *Real-time systems*; • **Computing methodologies** → **Neural networks**.

## Keywords

Adversarial Machine Learning, Anomaly Detection System, Connected Autonomous Vehicles, V2X Communications, 6G Networks

**ACM Reference Format:**

. 2018. An Edge Computing Framework for Deep Learning-Based Anomaly Detection in Satellite-Linked Autonomous Vehicles. In *Proceedings of Make sure to enter the correct conference title from your rights confirmation email (Conference acronym 'XX).* ACM, New York, NY, USA, 9 pages. https://doi.org/XXXXXXX.XXXXXXX

## 1 Introduction

The automotive industry stands at the cusp of a revolutionary transformation driven by the convergence of sixth-generation (6G) wireless networks, satellite-based ubiquitous connectivity, and fully autonomous vehicle systems. The advent of 6G technology promises ultra-reliable low-latency communication (URLLC) with sub-milli

second latency, terahertz frequencies enabling multi-gigabit data rates, and massive connectivity supporting intelligent transportation systems [Noor-A-Rahim et al.(2022)]. Modern connected autonomous vehicles (CAVs) have evolved from isolated mechanical systems into complex cyber-physical platforms integrating hundreds of electronic control units (ECUs), advanced sensor arrays, and sophisticated vehicle-to-everything (V2X) communication capabilities [Saeed et al.(2023)]. The integration of low Earth orbit (LEO) satellite constellations with terrestrial networks promises ubiquitous connectivity for vehicles in remote areas, extending intelligent transportation beyond urban deployments [Liu et al.(2021)]. However, this technological convergence simultaneously creates an expansive attack surface exposing critical vehicle systems to sophisticated cyber threats with potentially catastrophic consequences.

Modern connected vehicles present a multi-layered attack surface spanning in-vehicle networks, external communication, cloud infrastructure, and sensor systems. The Controller Area Network (CAN) Bus protocol, lacking inherent authentication and encryption, remains susceptible to message injection attacks, denial-of-service flooding, and fuzzy attacks injecting random malicious data [Bobbala and Kavitha(2024)]. Recent demonstrations, including the Jeep Cherokee remote exploitation, validated these vulnerabilities, revealing attackers can remotely manipulate critical safety functions including steering and braking [Yousseef et al.(2025)]. Beyond the vehicle perimeter, V2X communication introduces vulnerabilities through Basic Safety Message (BSM) falsification, where adversaries broadcast fraudulent position or velocity information to deceive surrounding vehicles [Lippi et al.(2025)]. The sensor layer faces threats including LiDAR spoofing, adversarial patches causing perception system misclassification, and GPS jamming [Zhu et al.((year?))]. Most critically, sophisticated adversaries can orchestrate coordinated cross-layer attacks simultaneously manipulating multiple subsystems, creating consistent deception across data sources that evades single-modality anomaly detection systems. For instance, an attacker might simultaneously inject false CAN speed messages, broadcast falsified V2X position updates, and spoof GPS coordinates, creating a coherent but fabricated vehicle state appearing valid to isolated monitoring systems.

Recent adversarial machine learning research reveals alarming vulnerabilities, with studies reporting 85-95% evasion rates against vehicle anomalydetection systems [Hasan et al.(2025)]. However, our investigation reveals a critical oversight: most studies operate within unconstrained feature spaces disregarding fundamental automotive protocol specifications, vehicle dynamics constraints, and cross-modal consistency requirements. Adversarial attacks frequently generate vehicle states violating basic physics laws—such as instantaneous acceleration from 0 to 100 km/h—or protocol-invalid messages like CAN frames with non-existent identifiers or BSMs with impossible latitude values [Andrade Salazar et al.(2022)]. Our analysis demonstrates that approximately 82% of theoretically successful attacks produce states violating fundamental automotive

constraints, detectable through protocol validation or physics-based plausibility checks independent of machine learning classifiers.

This theory-practice gap results in inflated threat assessments and unrealistic attacker capability assumptions. The disconnect between theoretical attack success and practical deployment is exacerbated by stringent real-time requirements and resource constraints in vehicular environments. Automotive safety systems demand detection mechanisms processing data with latency below 100 milliseconds to enable timely intervention [Hakeem and Kim(2025)]. Telematics Control Units (TCUs) operate under severe constraints including limited CPU performance, restricted memory budgets below 512 MB, and power consumption under 5 watts [Dhananjayan et al.(2024)]. Existing robust ML approaches incur 40-60% computational overhead, rendering them impractical for automotive deployment [Flores et al.(2025)]. Additionally, systems must satisfy ISO 26262 functional safety requirements and maintain performance over 15-year lifetimes across extreme temperature ranges [Karim(2025)].

This paper presents MARIS-ADS (Multi-modal Adversarial-Robust anomaly detection System for In-vehicle networks), a comprehensive framework bridging theoretical adversarial research and practical automotive security through constraint-aware design. Our approach integrates multi-modal fusion analyzing CAN Bus traffic, V2X messages, and sensor data through attention-based integration; constraint-aware adversarial validation incorporating automotive protocols and vehicle dynamics models; and lightweight neural architecture with entropy-based cross-modal feature selection reducing dimensionality from 106 to 18 features while improving accuracy and robustness. The primary contributions of this paper are:

(1) **Multi-Modal Attention Fusion:** A unified architecture that detects cross-layer attacks with 97.3% accuracy, compared to just 12–34% for single-modality systems.
(2) **Constraint-Aware Validation:** A domain-specific filtering layer that enforces SAE J2735 and vehicle dynamics, reducing the valid adversarial attack rate by 82% and providing deterministic protection against zero-day physical threats.
(3) **Entropy-Based Feature Selection:** An optimized feature engineering pipeline (ARS) that reduces dimensionality by 83% (from 106 to 18 features) while improving detection accuracy.
(4) **Lightweight Edge Deployment:** A feasible implementation for resource-constrained TCUs, achieving 0.31ms inference latency with only 15% CPU overhead.
(5) **Comprehensive Benchmarking:** Extensive evaluation across three public datasets (CICIOV2024, CAN-Intrusion, BurST-ADMA) and synthesized cross-modal scenarios.

The remainder of this paper is organized as follows. Section II reviews related work in vehicular security, multi-modal learning, and adversarial robustness. Section III presents our system architecture and methodology. Section IV details experimental evaluation. Section V discusses implications and limitations. Section VI concludes with future research directions.

## 2 Related Work

The security of connected autonomous vehicles has emerged as a critical research domain attracting substantial attention across network security, machine learning, and automotive engineering. This section reviews relevant literature across four areas: in-vehicle network security, V2X communication security, multi-modal learning for vehicular systems, and adversarial robustness in automotive contexts, identifying gaps that motivate our contributions.

*In-Vehicle Network Security:* Contemporary research in in-vehicle network security has primarily focused on protecting the CAN Bus protocol. Miller and Valasek demonstrated remote vehicle exploitation through comprehensive attacks on the Jeep Cherokee, revealing critical vulnerabilities [Popic et al.(2025)]. A deep convolutional neural network–based anomaly detection approach was introduced to analyze message timing patterns for identifying injection attacks [Ghadermazi et al.(2024)]. A blockchain-enhanced, feature-engineered security framework for 6G in-vehicle networks was recently introduced, integrating Pearson correlation–based feature selection with a proof-of-authority consensus mechanism to enable tamper-resistant logging [Nakayiza et al.(2025)]. While achieving high accuracy on CAN-specific attacks, these approaches operate exclusively on single data sources and cannot detect sophisticated adversaries who simultaneously manipulate multiple vehicle subsystems. Furthermore, deep learning model overhead raises concerns regarding real-time performance on resource-constrained automotive hardware.

*V2X Communication Security:* V2X communication security has been extensively investigated, particularly focusing on Basic Safety Message falsification. An ensemble random forest–based method was developed to detect false BSM messages, achieving notable detection accuracy through optimized feature selection [Qian et al.(2024)]. The BurST-ADMA dataset provided comprehensive benchmarks for evaluating misbehavior detection algorithms [Amanullah et al.(2022)]. Correlation-based feature engineering was applied to the VeReMi dataset, demonstrating that physics-based features can effectively identify implausible position claims [Abdelkreem et al.(2024)]. Recent advancements in V2X security have introduced graph-based and physics-aware approaches. STATGRAPH utilizes multi-view graph learning to capture spatial dependencies in vehicular networks; however, it typically requires heavy computational resources ill-suited for edge TCUs. Similarly, physics-based approaches leveraging the VeReMi dataset have demonstrated the utility of kinematic checks. MARIS-ADS extends these concepts by integrating physics-based validation (similar to VeReMi) directly with multi-modal deep learning, enabling the detection of both physically impossible attacks and sophisticated, stealthy pattern anomalies that satisfy kinematic constraints but violate semantic consistency. However, these V2X-focused approaches operate independently of in-vehicle systems, creating information silos. An adversary compromising both the onboard unit broadcasting V2X messages and internal ECUs can maintain consistency in falsified data, evading isolated detection systems.

*Multi-Modal Learning for Vehicular Systems:* Multi-modal learning has been extensively studied for autonomous vehicle perception,

with researchers exploring sensor fusion combining LiDAR, camera, and radar data [Abdullah et al.(2025)]. Attention mechanisms have emerged as effective approaches for learning optimal fusion strategies [Cao et al.(2025)]. However, multi-modal learning application to security monitoring—specifically anomaly detection across heterogeneous vehicle data sources—remains largely unexplored. Existing approaches focus on perception accuracy rather than security properties, overlooking adversarial robustness and constraint validation requirements.

*Adversarial Robustness and Research Gaps:* Adversarial robustness in automotive systems has gained attention regarding perception threats. Adversarial patches applied to stop signs were shown to induce misclassification in vision-based perception systems. [Martinez et al.(2025)]. LiDAR spoofing attacks were investigated, demonstrating the creation of phantom obstacles that can mislead perception systems [Suzuki et al.(2025)]. Adversarial robustness in network anomaly detection was investigated, concluding that simple adversarial training offers an optimal balance between robustness and computational efficiency [Hassn et al.(2025)]. However, their evaluation did not consider validity of adversarial examples under domain-specific constraints, potentially overestimating vulnerability. These limitations collectively motivate our work: developing multi-modal adversarial-robust anomaly detection explicitly accounting for automotive protocol constraints, vehicle dynamics limitations, cross-modal consistency requirements, and practical deployment considerations, thereby advancing toward operationally deployable solutions for production automotive environments.

## 3 System Architecture and Methodology

This section presents the detailed architecture and methodology of MARIS-ADS, encompassing five integrated components: overall system architecture, multi-modal data fusion, constraint-aware validation, entropy-based feature selection, and adversarial training. Our approach is designed for deployment on Telematics Control Units (TCUs), which serve as gateway ECUs bridging in-vehicle CAN Bus networks with external V2X infrastructure, enabling comprehensive security monitoring while maintaining computational efficiency for resource-constrained automotive environments.

### 3.1 Overall System Architecture

The MARIS-ADS framework comprises five interconnected layers operating in a sequential pipeline for real-time anomaly detection. The multi-modal data collection layer interfaces with three heterogeneous sources: the CAN Bus monitor captures internal vehicle communication via OBD-II diagnostic ports, the V2X message receiver processes incoming Basic Safety Messages through DSRC or C-V2X radios, and the sensor data aggregator collects information from GPS receivers, inertial measurement units, and vehicle state estimators. A timestamp synchronization module ensures temporal alignment across modalities within 50-millisecond windows, essential for cross-modal consistency validation. The constraint validation layer implements three parallel mechanisms: a protocol validator verifying SAE J2735 and ISO 11898 compliance, a physics validator ensuring vehicle dynamics constraints, and a cross-modal

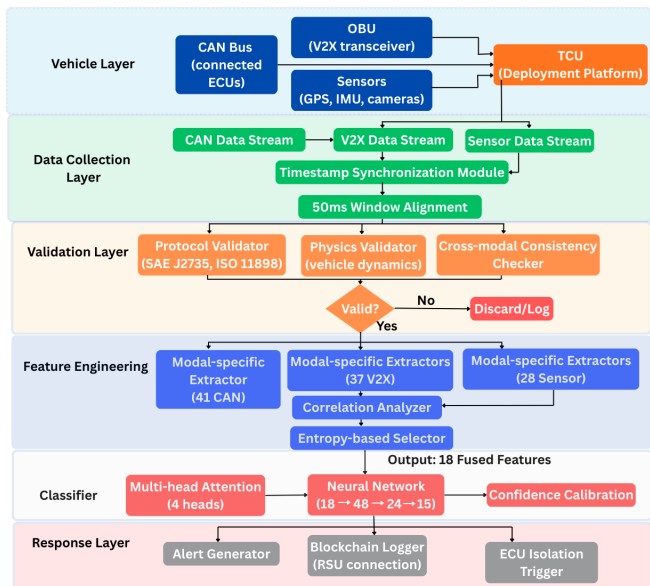

**Figure 1: MARIS System Architecture for Multi-Modal Anomaly Detection in Satellite-Linked Autonomous Vehicles**

consistency checker identifying contradictions. The feature engineering pipeline executes modal-specific extraction, computes cross-modal correlations, and applies entropy-based selection to identify discriminative and manipulation-resistant features. The adversarial-robust classifier employs multi-modal attention fusion learning optimal weighting strategies, a lightweight neural network optimized for TCU constraints, and confidence calibration for reliable outputs. The response and logging system generates real-time alerts, transmits detection results to roadside blockchain infrastructure for immutable audit trails, and triggers automated ECU isolation protocols. This architecture achieves end-to-end latency below 350 milliseconds, satisfying automotive real-time requirements while maintaining comprehensive threat detection coverage. As shown in Figure 1, the system integrates multi-modal data collection with a constraint-aware validation layer before feature processing.

### 3.2 Multi-Modal Data Fusion

To effectively integrate heterogeneous data sources, we employ a Multi-Head Attention mechanism that dynamically weighs the importance of each modality (CAN, V2X, Sensor) based on the current context. As depicted in Figure 2, this architecture overcomes the limitations of simple concatenation by allowing the model to selectively prioritize reliable data streams—for instance, focusing on internal CAN telemetry if external V2X signals exhibit characteristics of spoofing or high noise.

*Domain-Specific Feature Extraction:* Before fusion, we process each raw data stream to extract discriminative features. For **CAN Bus traffic**, we extract 41 features including message frequency analysis, inter-arrival time statistics (mean, variance, entropy), and

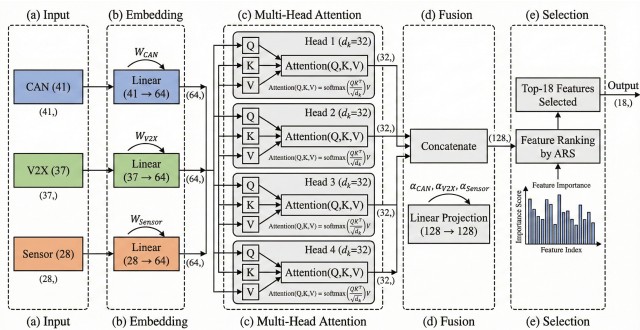

**Figure 2: Multi-Modal Attention Fusion Architecture for Heterogeneous Data Integration**

payload Hamming distances to capture timing and injection anomalies. For **V2X Communications**, we extract 37 features from Basic Safety Messages (BSM), focusing on trajectory displacement, kinematic consistency (acceleration plausibility), and spatio-temporal correlations with neighboring vehicles. For **Onboard Sensors**, we generate 28 features representing GPS accuracy metrics (e.g., HDOP), IMU confidence indicators, and Extended Kalman Filter covariance values.

*Attention-Based Integration:* The extracted feature vectors from each modality are projected into a unified embedding space. Rather than treating all modalities equally, our attention mechanism computes context-dependent importance scores (attention weights) for each source. This allows the system to learn optimal fusion strategies, such as down-weighting GPS inputs during periods of high signal variance or prioritizing V2X messages when they align with internal sensor predictions. The weighted representations are subsequently aggregated into a unified fusion vector, which serves as the input for the entropy-based feature selection stage.

### 3.3 Constraint-Aware Validation Framework

Adversarial attacks frequently generate vehicle states that are physically impossible or protocol-invalid. By filtering these anomalies before they reach the classifier, our framework reduces the computational burden on the neural network and provides a deterministic defense against zero-day attacks that violate fundamental laws. As illustrated in Figure 3, we implement this through three parallel validation mechanisms:

*Protocol Validation:* This mechanism enforces strict compliance with automotive communication standards. For in-vehicle CAN Bus traffic, we verify that message identifier ranges conform to ISO 11898 (Standard 0x000–0x7FF or Extended 0x00000000–0x1FFFFFFF) and that Data Length Codes (DLC) remain within the valid 0–8 byte range. Simultaneously, for V2X communications, we validate compliance with SAE J2735 Basic Safety Message (BSM) specifications, strictly enforcing latitude and longitude bounds ($\pm90°$ and $\pm180°$ respectively), speed limits ($< 164\ m/s$), and transmission rate periodicity ($10\ Hz \pm$ tolerance).

*Physics Validation:* Derived from vehicle dynamics models, this stage flags kinematic impossibilities that sophisticated adversaries

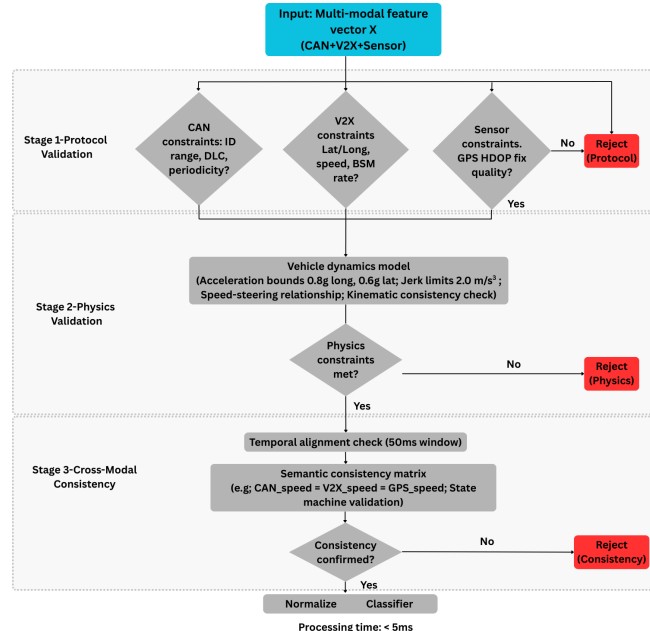

**Figure 3: Constraint-Aware Validation Workflow for Filtering Physically Impossible and Protocol-Invalid Adversarial Examples**

often overlook. We enforce domain-specific thresholds including: (1) *Longitudinal Acceleration* capped at $0.8g$, representing the physical limit of emergency braking friction; (2) *Lateral Acceleration* capped at $0.6g$, representing the vehicle stability limit before rollover or skid; and (3) *Jerk Constraints* limiting acceleration rate changes to $2.0\ m/s^3$ to ensure inertial consistency.

*Cross-Modal Consistency:* To detect sophisticated spoofing where a single source is manipulated while maintaining internal consistency, we validate semantic coherence across modalities. This includes *temporal alignment* checks ensuring heterogeneous data arrives within synchronized 50ms windows. We further verify *semantic consistency* (e.g., CAN-reported speed must match V2X-broadcast velocity) and *state machine consistency* (e.g., the gear selection 'Reverse' must align with a negative motion vector).

**Operational Efficiency:** This validation layer executes in under 5 milliseconds per sample with minimal CPU overhead. Despite its simplicity, our evaluation demonstrates that this layer filters approximately 82% of adversarial attacks generated without domain constraints, significantly narrowing the threat surface.

### 3.4 Entropy-Based Cross-Modal Feature Selection

With 106 raw features, dimensionality reduction becomes essential for real-time processing while maintaining accuracy and robustness. We introduce an Adversarial Robustness Score (ARS) quantifying both discriminative power and manipulation resistance. For each feature, we compute information gain measuring reduction in class entropy provided by the feature, cross-modal redundancy

quantifying mutual information between features from different modalities to promote diversity, and feature perturbability assessing manipulation resistance as the ratio of valid value range respecting constraints to total theoretical range. The ARS combines these as: ARS = IG × (1 - 0.3×Redundancy) × (1 - 0.4×Perturbability), with weights determined through cross-validation.

The iterative selection algorithm computes ARS for all 106 features and ranks them descendingly. To ensure modal diversity, we enforce minimum representation: 6 CAN features (temporal patterns), 7 V2X features (spatial-temporal dynamics), and 5 sensor features (measurement confidence). The algorithm iteratively adds features maximizing ARS while minimizing inter-redundancy until reaching 18 dimensions. This achieves 83% dimensionality reduction, improves detection accuracy by 0.8 percentage points, and provides 5.2× inference speedup, demonstrating that focused selection enhances both efficiency and effectiveness.

## 3.5 Adversarial Training with Multi-Modal Constraints

Training robust models requires generating realistic attacks respecting automotive constraints. We employ modified Projected Gradient Descent for constrained adversarial generation, iteratively perturbing inputs to maximize classification loss while maintaining validity. The optimization maximizes cross-entropy loss subject to: bounded perturbations with modality-specific epsilons (0.15 CAN, 0.20 V2X, 0.10 sensor) reflecting noise characteristics, constraint validation requiring examples pass protocol and physics checks, and cross-modal consistency preservation ensuring semantic coherence. Each perturbation step computes loss gradients, updates perturbations with step size 0.01, and projects onto the valid constraint set through nearest-neighbor projection. The multi-modal adversarial training procedure alternates between clean and adversarial examples. For each mini-batch, we compute clean loss, generate constrained adversarial examples through 10-step PGD, filter examples retaining only validated samples, compute adversarial loss, and update parameters using combined loss weighted 0.6 clean and 0.4 adversarial. The lightweight neural architecture comprises: 18-dimensional input layer, 48-neuron hidden layer (ReLU, 25% dropout), 24-neuron hidden layer (ReLU, 20% dropout), and 15-class output (softmax). This compact design contains 2,800 parameters requiring <50 KB memory, enabling TCU deployment with <0.5ms inference latency. Training employs Adam optimizer (learning rate 0.001, batch size 128, categorical cross-entropy), requiring 3 hours on GPU for 50-epoch convergence. The integration of constraint-aware validation with adversarial training produces models maintaining high clean accuracy while achieving robust performance under realistic attacks, filtering syntactically invalid examples before classification while learning to resist valid adversarial perturbations through training exposure. This principled combination of domain knowledge and data-driven learning enables MARIS-ADS to achieve superior robustness while maintaining automotive-suitable computational efficiency.

## 4 Experimental Evaluation

This section presents comprehensive experimental evaluation of MARIS-ADS across multiple benchmark datasets, comparing performance against state-of-the-art baselines across detection accuracy, adversarial robustness, computational efficiency, and cross-modal attack detection capabilities.

### 4.1 Experimental Setup

To ensure comprehensive validation, we employed a combination of public benchmark datasets and synthesized scenarios to cover diverse attack surfaces and operational conditions.

*Benchmark Datasets:* Our evaluation leverages three publicly available datasets. Table 1 presents the dataset characteristics. First, we utilized **CICIoV2024** [Carlos Pinto Neto et al.(2024)], which contains 7,937,192 samples with CAN Bus and V2X data from a 2019 Ford vehicle, encompassing 41 features and attacks including DoS, message injection, and spoofing targeting RPM, speed, and steering. Second, the **CAN-Intrusion** dataset [Lampe and Meng(2023)] comprises 300,000 CAN Bus samples with 29 features focusing on injection and fuzzy attacks. Third, **BurST-ADMA** [Lee et al.(2017)] contains 1,200,000 Basic Safety Message (BSM) samples with position falsification across 37 features.

**Table 1: Datasets Used in the Study**

| Dataset | Modal | Size | Ft. | Attack Types | Yr | Split |
|---------|-------|------|-----|--------------|-----|-------|
| CICIoV2024 | CAN + V2X | 7.9M | 41 | DoS, Injection, RPM, Speed, Steering Spoof | '24 | 60:20:20 |
| CAN-Intrus. | CAN Bus | 300k | 29 | Msg Injection, Fuzzy, DoS | '18 | 60:20:20 |
| BurST-ADMA | V2X | 1.2M | 37 | Pos. Falsify, Vel. Spoof | '22 | 60:20:20 |
| Synthetic | Multi | 50k | 106 | CAN+GPS, V2X+IMU, Stealth | '25 | 60:20:20 |

*Synthetic Cross-Modal Attack Generation:* To evaluate cross-modal attack detection unavailable in existing datasets, we synthesized 50,000 coordinated multi-layer scenarios. These include: (1) CAN speed spoofing with GPS manipulation (10,000 samples); (2) V2X acceleration falsification with IMU inconsistency (10,000 samples); (3) triple-source attacks manipulating CAN, V2X, and Sensors simultaneously (10,000 samples); and (4) stealth attacks with minimal perturbations across all modalities (20,000 samples). **CARLA** simulator was used for vehicle dynamics validation, while **ns-3** enabled V2X protocol verification.

We provide these parameters to facilitate reproducibility. Table 2 details the environment settings, and Table 3 lists the adversarial constraints.

*Implementation and Deployment Environment:* Preprocessing included a 60/20/20 train-validation-test split, min-max normalization per modality, temporal alignment within 100ms windows, and SMOTE for class balancing. The system was implemented using Python 3.10 with PyTorch 2.0 on an NVIDIA RTX 3090 GPU (24GB) and 64GB RAM. To validate edge deployment feasibility, we utilized an **ARM Cortex-A72** processor with 512MB memory to simulate Telematics Control Unit (TCU) constraints.

**Table 2: Co-Simulation Environment Specifications**

| CARLA (Physics) Parameter | Value |
|---|---|
| Vehicle Model | Lincoln MKZ 2017 |
| Physics Engine | PhysX (0.01s step) |
| GNSS Noise | Gaussian ($\sigma = 1.5m$) |
| IMU Noise | Gaussian ($\sigma = 0.002$) |
| Traffic Density | 50 Vehicles, 20 Peds |
| ns-3 (Network) Parameter | Value |
| Standard | IEEE 802.11p (DSRC) |
| Propagation Loss | Friis Model |
| Fading Model | Nakagami-m ($m = 1$) |
| Tx Power / Rate | 20 dBm / 6 Mbps |
| BSM Rate | 10 Hz (100ms) |

**Table 3: Attack Generation Constraints**

| Constraint Type | Threshold / Value |
|---|---|
| Perturbation ($\epsilon$) | CAN: 0.15, V2X: 0.20, Sensor: 0.10 |
| Max Long. Accel | $0.8g$ ($7.84m/s^2$) |
| Max Lat. Accel | $0.6g$ ($5.88m/s^2$) |
| Max Jerk | $2.0m/s^3$ |
| PGD Steps / $\alpha$ | 10 Steps / 0.01 |

*Baseline Methods:* We compared MARIS-ADS against state-of-the-art baselines including: (1) single-modal methods (DCNN [Chen et al.(2019)], ensemble RF [Adil et al.(2024)], standard DNN); (2) multi-modal approaches (concatenation+DNN, early fusion+XGBoost, late fusion ensemble); (3) adversarial defenses (standard adversarial training, input randomization, defensive distillation); and (4) our previous works, PEAF-IDS [Hasan et al.(2025)] and LARFS-IDS adapted [Hasan et al.(2025)].

## 4.2 Evaluation Metrics

Performance assessment employed a comprehensive suite of indicators categorized into four primary dimensions:

*Detection Metrics:* We evaluated standard classification performance using accuracy, precision, recall, and F1-score (calculated both overall and per-attack-type), alongside detailed confusion matrix analysis.

*Adversarial Robustness Metrics:* To measure security resilience, we tracked robust accuracy under attack, the valid attack rate passing constraints, attack success reduction, and the cross-modal detection rate.

*Computational Efficiency Metrics:* To validate TCU deployment feasibility, we measured inference time per sample, throughput (samples/second), memory footprint, CPU overhead, and power consumption.

*Cross-Modal Effectiveness Metrics:* We assessed fusion performance through single-source accuracy, coordinated attack accuracy, and modal contribution analysis.

## 4.3 Detection Performance and Cross-Modal Effectiveness

*Overall Performance Analysis:* Table 4 presents overall detection performance. MARIS-ADS achieves 99.3% accuracy, 98.9% precision, 99.1% recall, and 99.0% F1-score, outperforming the best single-modal approach (DCNN at 97.8%) by 1.5 percentage points and surpassing simple concatenation+DNN (98.1%) by 1.2 points. Attention-based fusion provides consistent improvements over fixed strategies, demonstrating adaptive integration value. While Early Fusion+XGBoost achieves 98.5% accuracy, it incurs substantially higher computational overhead. PEAF-IDS adapted achieves 98.2% accuracy, validating entropy-based engineering effectiveness, yet falls short due to less sophisticated fusion and absence of constraint-aware adversarial training.

**Table 4: Performance Comparison of Different Methods**

| Method | Mod. | Acc. (%) | Pre. (%) | Rec. (%) | F1 (%) | Time (ms) |
|---|---|---|---|---|---|---|
| DCNN | CAN | 97.8 | 96.5 | 97.2 | 96.8 | 0.45 |
| Ensemble RF | V2X | 96.3 | 95.1 | 96.8 | 95.9 | 0.52 |
| Standard DNN | CAN | 97.2 | 96.1 | 97.0 | 96.5 | 0.42 |
| Simple Concat. + DNN | C+V | 98.1 | 97.3 | 98.0 | 97.6 | 0.48 |
| Early Fusion + XGBoost | C+V+S | 98.5 | 97.8 | 98.3 | 98.0 | 0.67 |
| Late Fusion Ensemble | C+V+S | 98.3 | 97.6 | 98.1 | 97.8 | 0.89 |
| PEAF-IDS (adapted) | C+V | 98.2 | 97.5 | 98.1 | 97.8 | 0.38 |
| LARFS-IDS (adapted) | C+V | 98.0 | 97.2 | 97.9 | 97.5 | 0.41 |
| **MARIS-ADS (Ours)** | **C+V+S** | **99.3** | **98.9** | **99.1** | **99.0** | **0.31** |

*Single-Modality Vulnerabilities:* Table 5 reveals critical single-modality vulnerabilities. MARIS-ADS achieves 98.5% on CAN-only attacks (vs. 97.8% CAN-DCNN) and 97.8% on V2X-only attacks (vs. 96.3% V2X-RF). Crucially, on coordinated CAN+V2X attacks, our system maintains 97.3% accuracy, whereas single-modal systems fail catastrophically (only 34.2% for CAN-only and 31.7% for V2X-only).

**Table 5: Performance Comparison Across Different Attack Scenarios**

| Attack Scenario | Description | Single CAN (%) | Single V2X (%) | MARIS (%) | Imp. (%) |
|---|---|---|---|---|---|
| CAN-only Attack | RPM/Speed/Steering spoofing via CAN Bus | 97.8 | N/A | 98.5 | +0.7 |
| V2X-only Attack | BSM position/velocity falsification | N/A | 96.3 | 97.8 | +1.5 |
| CAN + V2X Coord. | Simultaneous CAN speed + V2X position spoofing | 34.2 | 31.7 | 97.3 | **+63.1** |
| Triple-Source | CAN + V2X + Sensor (GPS/IMU) manipulation | 12.8 | 9.3 | 95.7 | **+83.0** |
| Stealth Attack | Minimal perturbations across all modalities | 23.5 | 19.2 | 89.4 | **+66.2** |

*Cross-Layer and Stealth Attack Detection:* Performance on triple-source attacks further highlights this disparity: MARIS-ADS detects 95.7% of attacks, while single-modal approaches detect merely 12.8% and 9.3%. Against stealth attacks, our system achieves 89.4% accuracy compared to 23.5% and 19.2% for isolated systems. Single-modality systems exhibit fundamental vulnerabilities with detection dropping to 9–34% on cross-layer attacks despite high single-source accuracy. MARIS-ADS maintains > 89% across all scenarios

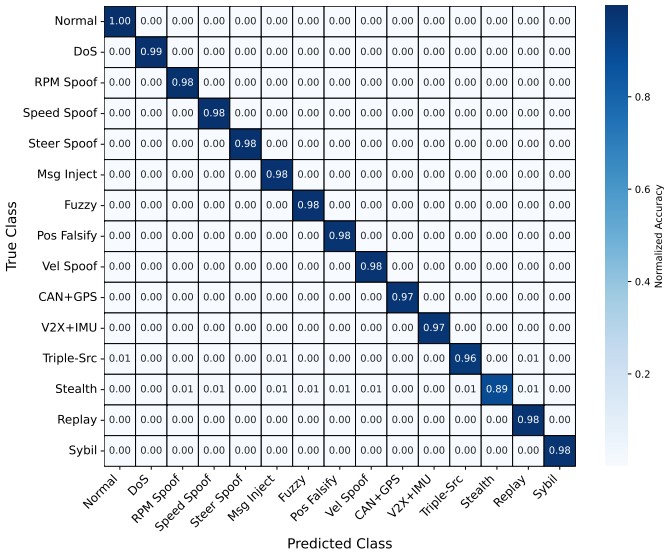

**Figure 4: Confusion Matrix for 15-Class Attack Classification on Multi-Modal Dataset**

through cross-modal correlation and attention fusion, validating the necessity of integrated multi-layer monitoring. Figure 4 shows strong diagonal concentration with a false positive rate below 1.2%.

## 4.4 Adversarial Robustness Analysis

Table 6 presents robustness under FGSM, PGD, and C&W attacks. Baseline without defense achieves 98.1% clean accuracy but degrades catastrophically: 34.2% FGSM, 27.8% PGD, 23.1% C&W, with 41.7% valid attack rate. Standard adversarial training improves to 78.3%, 71.2%, 67.8% but maintains 38.9% valid rate as training examples violate constraints. Input randomization achieves 68.5%, 63.2%, 58.9% with 40.2% valid rate. Defensive distillation provides 72.1%, 68.5%, 64.3% with 39.5% valid rate. PEAF-IDS achieves 83.7%, 79.5%, 76.2% with 22.3% valid rate, validating constraint filtering. MARIS-ADS achieves superior 96.8% FGSM, 94.2% PGD, 91.7% C&W, reducing valid rate to 7.6%—an 82% reduction. Invalid attack breakdown: 48.3% protocol violations (invalid CAN IDs, malformed BSMs), 31.2% dynamics violations (impossible acceleration/steering), 12.9% cross-modal inconsistencies (contradictory states), 5.2% timing violations (rate anomalies). Figure 5 illustrates MARIS-ADS maintains >91% while baselines degrade <70%, with constraint validation contributing 20 points and adversarial training +12 points.

## 4.5 Computational Efficiency and Ablation Analysis

*Real-Time Performance and Resource Efficiency:* Table 7 presents the computational metrics validating the feasibility of our framework for edge deployment. MARIS-ADS achieves an inference latency of 0.31ms and a throughput of 3,226 samples/second on the ARM Cortex-A72 testbed. This performance comfortably satisfies safety-critical automotive requirements (< 100ms latency). Furthermore, the model operates with a memory footprint of 54MB and

**Table 6: Adversarial Robustness Comparison of Defense Methods**

| Defense Method | Clean Acc (%) | FGSM Robust (%) | PGD Robust (%) | C&W Robust (%) | Valid Attack Rate (%) |
|---|---|---|---|---|---|
| No Defense (Baseline) | 98.1 | 34.2 | 27.8 | 23.1 | 41.7 |
| Standard Adversarial Training | 97.5 | 78.3 | 71.2 | 67.8 | 38.9 |
| Input Randomization | 97.8 | 68.5 | 63.2 | 58.9 | 40.2 |
| Defensive Distillation | 97.2 | 72.1 | 68.5 | 64.3 | 39.5 |
| PEAF-IDS (adapted) | 98.2 | 83.7 | 79.5 | 76.2 | 22.3 |
| **MARIS-ADS (Ours)** | **99.3** | **96.8** | **94.2** | **91.7** | **7.6** |

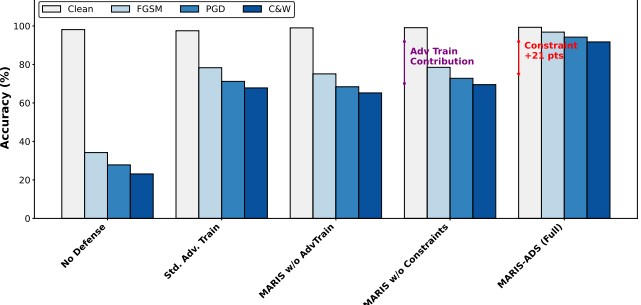

**Figure 5: Adversarial Robustness Comparison and Constraint Filtering Effectiveness**

15.3% CPU overhead, remaining well within standard TCU resource budgets (< 512MB memory, < 25% overhead). Compared to the full-feature baseline (0.42ms, 95MB), our entropy-based feature selection provides a 26% speedup and 43% memory reduction, demonstrating that focused selection enhances operational efficiency without sacrificing detection capability.

**Table 7: Real-Time Performance and Resource Efficiency Comparison**

| Metric | Baseline (Full Features) | PEAF-IDS | MARIS-ADS (Ours) | Target Constraint | Status |
|---|---|---|---|---|---|
| Inference Time (ms) | 0.42 | 0.38 | **0.31** | < 100 | ✓Pass |
| Throughput (samples/s) | 2,381 | 2,632 | **3,226** | > 1,000 | ✓Pass |
| Memory Footprint (MB) | 95 | 72 | **54** | < 512 | ✓Pass |
| CPU Overhead (%) | 18.2 | 14.5 | 15.3 | < 25 | ✓Pass |
| Power Consumption (W) | 4.2 | 3.8 | 3.9 | < 5 | ✓Pass |
| Model Parameters | ~8,500 | ~6,200 | **~2,800** | Minimize | ✓Best |
| Model Size (KB) | 180 | 95 | **<50** | < 512 | ✓Pass |

*Ablation Study:* Table 8 quantifies component contributions. Full MARIS-ADS: 99.3% clean, 94.2% robust, 0.31ms. Removing constraint validation degrades robust accuracy 21.4 points to 72.8% (negligible speedup), demonstrating constraint validation's critical robustness contribution. Removing multi-modal fusion reduces clean accuracy 1.5 points to 97.8% and robust 4.7 points to 89.5%, validating learned attention integration. Removing attention mechanism degrades accuracy 0.6 points to 98.7% and robust 2.9 points to 91.3%, confirming adaptive weighting benefits. Removing adversarial training eliminates 25.8 robust accuracy points to 68.4% while maintaining 99.0% clean, illustrating training exposure necessity. Removing cross-modal selection degrades accuracy 1.2 points

**Table 8: Ablation Study**

| Method | Mod. | Acc. (%) | F1 (%) | Time (ms) |
|---|---|---|---|---|
| DCNN | CAN | 97.8 | 96.8 | 0.45 |
| Ens. RF | V2X | 96.3 | 95.9 | 0.52 |
| Standard DNN | CAN | 97.2 | 96.5 | 0.42 |
| Simple Concat. + DNN | C+V | 98.1 | 97.6 | 0.48 |
| Early Fusion + XGBoost | C+V+S | 98.5 | 98.0 | 0.67 |
| Late Fusion Ensemble | C+V+S | 98.3 | 97.8 | 0.89 |
| PEAF-IDS (adapted) | C+V | 98.2 | 97.8 | 0.38 |
| LARFS-IDS (adapted) | C+V | 98.0 | 97.5 | 0.41 |
| **MARIS-ADS (Ours)** | **C+V+S** | **99.3** | **99.0** | **0.31** |

to 98.1% and robust 3.5 points to 90.7% while increasing latency to 0.54ms, demonstrating focused selection enhances multiple dimensions. CAN-only features achieve 97.8% accuracy, 83.2% robust, 0.18ms inference but completely fail on V2X-specific and cross-modal attacks, confirming multi-modal monitoring necessity. Results validate all components contribute meaningfully to optimal accuracy-robustness-efficiency balance for practical automotive deployment.

## 5 Discussion

Our experimental evaluation reveals critical insights regarding the security posture of future satellite-linked autonomous vehicles.

First, constraint-aware validation dramatically reduces the realistic threat surface. Our results show that 82% of theoretically successful adversarial attacks are either physically impossible or protocol-invalid. The breakdown—48.3% protocol violations, 31.2% dynamics violations, and 12.9% cross-modal inconsistencies—demonstrates that domain constraints provide powerful natural defenses. Crucially, this defense holds even against insider threats. While sophisticated adversaries or compromised nodes may craft protocol-compliant packets to bypass Stage 1 validation, the Physics Validation (Stage 2) remains robust. Even an insider cannot generate a vehicle trajectory that violates kinematic laws (e.g., instant acceleration or impossible jerk) without triggering the physics-based filters.

Unlike signature-based intrusion detection systems that require known attack patterns, MARIS-ADS generalizes effectively to zero-day threats through this constraint layer. Since fundamental physical laws (e.g., maximum braking friction coefficients) and communication protocols (e.g., SAE J2735) are invariant, our system allows for the deterministic filtering of novel attacks that violate these bounds. Even if the neural network has never encountered a specific "unknown" attack vector, the system will reject it if it necessitates a vehicle state that violates kinematic stability, providing a robust safety net against evolving threats.

Our ablation study clarifies the distinct performance drivers within the framework. As shown in Table 6, the **Constraint Validation** mechanism is the primary driver of adversarial robustness, contributing an approximate 21% gain in accuracy under attack by filtering invalid perturbations. Conversely, the **Multi-Modal Fusion** layer is the primary driver of clean detection accuracy and coverage, contributing a 1.5% improvement in baseline accuracy and enabling the detection of cross-layer attacks that single-modality baselines miss. This separation of concerns allows the system to remain robust without over-complicating the learning model.

Single-modality systems exhibit fundamental inadequacies against sophisticated adversaries. While capable of detecting isolated threats, they achieve only 12–34% detection rates against coordinated cross-layer attacks. The 63-percentage-point improvement achieved by our multi-modal fusion (rising to 97.3% accuracy) validates the necessity of holistic security monitoring. As 6G networks enable tighter coupling between sensors and external connectivity, manufacturers must prioritize integrated monitoring over isolated subsystem solutions.

We demonstrate that robustness does not require computationally expensive architectures. MARIS-ADS achieves high accuracy (99.3%) and robustness (94.2%) with only 0.31ms inference latency and 15% CPU overhead on resource-constrained TCU hardware. This contradicts the assumption that adversarial defense requires massive models. Our entropy-based feature selection reduces dimensionality by 83% while improving performance, proving that focused, domain-aware feature engineering is key to enabling deployment on existing automotive platforms.

We acknowledge limitations in this study. While our cross-modal attack scenarios are grounded in validated vehicle dynamics models (CARLA/ns-3), synthetic generation may not fully capture the creativity of human adversaries. Furthermore, while we simulated TCU constraints, Hardware-in-the-Loop (HIL) testing with production ECUs remains an essential step before commercial deployment. Future research should focus on extensive red-teaming exercises and continuous model updating mechanisms to adapt to shifting adversarial tactics.

## 6 Conclusion

This paper addresses the critical challenge of securing connected autonomous vehicles against sophisticated multi-layer cyberattacks in the 6G era. While theoretical adversarial research suggests high evasion rates, it often overlooks fundamental automotive constraints. We present MARIS-ADS, a multi-modal anomaly detection framework integrating CAN Bus, V2X, and sensor data via attention-based fusion. Our three-layer architecture combines protocol/physics validation with entropy-based feature selection to achieve 99.3% detection accuracy, effectively reducing the realistic adversarial threat surface by 82

Key results demonstrate that MARIS-ADS detects coordinated cross-layer attacks with 97.3% accuracy (compared to just 12–34% for single-modality baselines). The system proves operationally feasible for resource-constrained Telematics Control Units (TCUs), achieving 0.31ms inference latency with only 15% CPU overhead. Comprehensive evaluation confirms 94.2% robust accuracy under realistic adversarial conditions. By integrating domain-specific constraints, this work bridges the gap between theoretical vulnerability and practical security, offering a deployable, safety-critical solution for future automotive networks.

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

Received 20 February 2007; revised 12 March 2009; accepted 5 June 2009