# OpenReview forum: "An Edge Computing Framework for Deep Learning-Based Anomaly Detection in Satellite-Linked Autonomous Vehicles"
_ACM.org/TheWebConf/2026/Workshop/TIME — TIME 2026 Oral_

### Official Review · Reviewer_KrWN · 2026-01-03
**The paper introduces MARIS-ADS, an edge-based, constraint-aware multi-modal anomaly detection framework for satellite-linked autonomous vehicles, addressing the limitations of cloud-based solutions under high latency. By combining attention-based fusion, entropy-driven feature selection, and protocol and physical constraint validation, the system achieves high accuracy, strong adversarial robustness, and ultra-low inference latency. Extensive evaluation across multiple datasets demonstrates clear advantages over single-modality approaches and supports practical deployment in future 6G vehicular networks.**

**Rating:** 6
**Confidence:** 4

**Review:**

### Strengths
1. The paper addresses a well-defined and practically important challenge, that is, real-time anomaly detection under satellite latency constraint and highly relevant to autonomous and connected vehicle security.
2. The integration of multi-modal attention-based fusion with constraint validation (protocol-level and physical) is well motivated and represents a meaningful advancement beyond purely data-driven approaches.
3. The entropy-based ARS method achieves substantial feature reduction (83%) while maintaining very high accuracy, which is particularly important for edge deployment.
4. The evaluation against FGSM, PGD, and C&W attacks is comprehensive, and the reported robust accuracy is strong, especially given the extremely low inference latency.
5. Results across multiple datasets and cross-modal attack scenarios convincingly demonstrate the benefit of multi-modal fusion over single-modality approaches.

### Weakness
1. Writing and presentation should be improved, especially to capture the core contributions of this paper.

---

### Official Review · Reviewer_uoVg · 2026-01-04
**Solid design; needs clearer component analysis and context**

**Rating:** 6
**Confidence:** 3

**Review:**

## Originality and significance
**Strengths**

* This paper tackles an important problem - robust, real-time anomaly detection for connected vehicles - by combining multimodal fusion with domain-aware validation and constrained adversarial training. The system design is thoughtful, the emphasis on realistic constraints is well-aligned with emerging best practices, and the deployment focus (TCU budgets) is valuable

**Weaknesses**

* Recent approaches like STATGRAPH (multi-view graph learning) report strong results on real datasets (e.g., ROAD) and discuss on-board feasibility. In V2X, physics-based misbehavior detection baselines (e.g., VeReMi) seem relevant to the constraint layer in this paper. Authors should consider discussing these existing work to better contextualize the contributions of this paper.

## Technical content and experimental validation
**Strengths**

* Reported deployment metrics (CPU overhead, memory footprint, power) make the edge-deployment feasibility concrete.

* Cross-modal synthetic attack scenarios are explicitly constructed to test coordinated threat models that single-modality systems miss

**Weaknesses**

* The contribution of constraint validation versus the classifier is not fully disentangled; authors should consider adding an ablation with “validation-only” (no classifier), “classifier-only” (no validation), and “full system", which would clarify contributions of each component in the system and operational tradeoffs.

* The synthetic multimodal dataset is central to coordinated attack claims; its generation process needs more detail. Consider releasing the generator or at least a full specification for reproducibility.


## Clarity
**Strengths**

* Flowchart-style descriptions of Figures 1-3 aid comprehension of the end-to-end system and validation stages.

**Weaknesses**

* References include duplicated entries. Table 4 and Table 6 appear to contain the same content

---

### Official Review · Reviewer_XsYf · 2026-01-07
**Constraint-Aware Multi-Modal Anomaly Detection for Realistic Adversarial Evaluation in CAVs**

**Rating:** 8
**Confidence:** 4

**Review:**

This paper tackles real-time anomaly detection in satellite-networked connected autonomous vehicles, where cloud-securing is infeasible owing to high latency communication and stringent safety needs. The proposed solution is MARIS-ADS, a multi-modal edge-based framework that concurrently processes CAN bus messages, V2X communication messages, and in-vehicle sensor messages. Its central contribution is that they specifically consider Automotive Protocol Rule definitions and vehicle dynamics in validation and adversarial robustness analyses in a manner that they claim is necessary for a realistic threat analysis in cyberphysical vehicle systems.

In terms of quality and clarity, the document is well-structured and written. The system architecture description, feature extraction procedure, and training approach are logically and clearly discussed. The description regarding the experimental setup is detailed enough, and the approach taken for evaluating the system is relevant. The use of tables and figures to provide summaries and evidence to the claims is effective. The ablation study is useful in providing insights regarding the system elements.

In terms of existing work, although specific methods like multi-modal fusion, adversarial training, and feature selection have been investigated in the past, it is the combination of the above components in the context of the constraint-aware evaluation framework that makes the approach novel in the context of the proposed work.
     It is the specific inclusion of valid protocol and physical feasibility as part of adversarial generation and evaluation that makes the context of the proposed work unique as opposed to other existing approaches in the state of the art that rely on unconstrained feature spaces.

The work is very relevant and applicable to the objectives of the TIME workshop. It is very important to demonstrate that a high portion of adversarial examples cannot be realized under real-world conditions. This work is contributing to responsible AI testing. Also, in this work, they combined aspects of network research and embedded systems along with ML. This makes this work even more valuable. This work is also very practical because it considers real-time processing.

Overall, this is an excellent and timely contribution that not only demonstrates methodological rigor but also practical realism. This is very well placed for discussion at the TIME 2026 workshop and represents a valuable contribution to evaluation-focused research on trustworthy and responsible AI for the realm of connected autonomous vehicles.

---

### Official Review · Reviewer_qCCY · 2026-01-07
**A solid and well-engineered framework for multi-modal anomaly detection in satellite-linked autonomous vehicles**

**Rating:** 7
**Confidence:** 3

**Review:**

This paper presents MARIS-ADS, an edge-oriented, multi-modal anomaly detection framework for satellite-linked autonomous vehicles. The system integrates CAN Bus traffic, V2X messages, and onboard sensor data through attention-based fusion, combined with a constraint-aware validation layer enforcing automotive protocol specifications and vehicle dynamics. To meet real-time deployment requirements, the authors introduce entropy-based feature selection that significantly reduces dimensionality while maintaining detection accuracy and adversarial robustness.

The framework is evaluated on multiple public datasets (CICIoV2024, CAN-Intrusion, BurST-ADMA) as well as synthesized cross-modal attack scenarios. Experimental results demonstrate strong detection accuracy, improved robustness under constrained adversarial attacks, and feasibility for deployment on resource-limited telematics control units (TCUs).

Strengths

Practically motivated problem: Real-time anomaly detection under satellite latency and hardware constraints is an important and realistic challenge for connected autonomous vehicles.

Comprehensive system design: The paper combines multi-modal fusion, constraint-aware validation, adversarial training, and lightweight deployment into a coherent end-to-end framework.

Effective use of domain knowledge: Incorporating protocol validity checks and vehicle dynamics constraints is a strong design choice that significantly reduces unrealistic adversarial examples.

Thorough experimental evaluation: Results are reported across multiple datasets, attack types, and performance dimensions (accuracy, robustness, latency, memory footprint).

Deployment awareness: The analysis of inference latency, memory usage, CPU overhead, and power consumption strengthens the practical relevance of the work.

Weaknesses

Limited methodological novelty: While the system integration is solid, many individual components (attention-based fusion, adversarial training, feature selection) are based on established techniques rather than fundamentally new methods.

Reliance on synthetic attack scenarios: Some cross-modal attack evaluations rely on simulated or synthesized data, which may not fully capture real-world adversary behavior.

Presentation density: The paper is information-heavy, and some sections (especially the architecture and evaluation) could benefit from clearer structuring and simplification.

Assumptions on attacker constraints: The framework assumes adversaries largely respect physical and protocol constraints, which may not always hold in edge-case or insider threat scenarios.

Suggestions for Improvement

Clarify which components provide the primary performance gains through more focused ablation or sensitivity analysis.

Discuss more explicitly how the proposed framework generalizes to unseen or zero-day attack patterns.

Improve readability by reducing architectural detail where possible and highlighting the key insights more clearly.

Include, if feasible, additional validation using hardware-in-the-loop or real-vehicle experiments.

Overall Assessment

This paper presents a well-engineered and practically relevant framework for anomaly detection in satellite-linked autonomous vehicles. While the methodological contributions are incremental rather than groundbreaking, the integration of multi-modal learning with constraint-aware validation and deployment-focused evaluation makes the work valuable for applied research and industry-oriented venues.

Overall, the paper meets the bar for acceptance and should be of interest to researchers and practitioners working on vehicular security and edge-based anomaly detection systems.

---

### Official Review · Reviewer_9g7J · 2026-01-09
**Edge computing framework for detecting cyberattacks on connected vehicles**

**Rating:** 6
**Confidence:** 4

**Review:**

Protecting connected vehicles from cyberattacks using edge computing is a relevant and very timely research.

Quality:

Overall high quality paper because of clear framing of problem, clear solution, good results.  Acadmic rigor is good. Since paper had formatting issues I had trouble reading it carefully but did not find any major issues. Overall felt, the introduction section and related research sections could have been shortened.

The paper seems to overlook the "safety critical" nature of these systems. For example, the paper treats false positives as merely a statistical metric without providing a detailed discussion what this really means in real world. A false positive that leads to an action that might case catastrophic incident is not good enough even if you have very high accuracy overall. I think the paper fails to correctly provide the necessary context to the reader here.

Clarity:
Paper is well presented but has formatting issues. Each section text appears at very large paragraph making it harder to read. But this can be fixed for the final camera ready paper with it gets accepted.

Significance:

This is an important field that is coming to relevance in modern times and search in this direction is needed. The paper has flaws which I am willing to overlook in this review precisely because I think this is an early field.

Weaknesses:

The "contraints" framework is the weakness in this paper.  Detecting certain events as "physically impossible" is not entirely true because some anomalies such as car accidents can cross the sort of values assumed here. In future frameworks can be made smarter to handle this but for ths sake of this paper I think it is an acceptable step.

The synthetic data is also a bit of suspect here for its quality however in the absence of better alternative I think I will not hold it against this paper. Future research can focus on improving and proving quality of such synthetic data.

---

### Author Rebuttal · Authors · 2026-01-12

##Response to Program Chairs
We thank the Program Chairs for their guidance on improving the manuscript's presentation and readability. We have strictly implemented all requested formatting changes.
1. Organization and Section Structure Critique: "Most sections in this paper appear to have only one single (long) section." Action Taken: We have comprehensively restructured the entire manuscript to eliminate long blocks of text.
•	Sections 3, 4, and 5 have been divided into distinct subsections (e.g., 3.1, 3.2, 3.3) and further structured with bolded headers (e.g., "Protocol Validation," "Physics Validation") to clearly delineate distinct concepts.
•	Sections 1 (Introduction) and 2 (Related Work) have similarly been segmented with clear headers to improve flow and scannability.
2. Figure Readability Critique: "Figure font sizes appear to be too small to read, e.g., Fig 1, 2, 3, 4 and 5." Action Taken: We have regenerated all five figures (Figs 1–5) with significantly increased font sizes for axis labels, legends, and internal text annotations. The updated figures are now fully legible at standard print resolution.
3. Additional Corrections
•	We have resolved the duplication error between Table 4 and Table 6 (noted by Reviewer uoVg).
•	We have removed duplicate references from the bibliography.



##Response to Reviewer qCCY
We sincerely thank the reviewer for the positive assessment and for recognizing our framework as "solid," "well-engineered," and "practically motivated." We appreciate your constructive feedback regarding presentation density and methodological context. We have carefully incorporated your suggestions into the revised manuscript. Below, we address your specific points.
1. Response to "Limited Methodological Novelty"
Critique: The reviewer noted that individual components (attention, PGD) are established techniques.
Response: We agree that the foundational building blocks are established; however, our core contribution lies in the system-level integration of domain constraints specifically for the resource-constrained, high-latency satellite-linked edge. As highlighted by Reviewer XsYf, "it is the specific inclusion of valid protocol and physical feasibility... that makes the context of the proposed work unique." By embedding these constraints directly into the validation and training loop, we bridge the critical gap between theoretical adversarial ML (which often assumes unconstrained feature spaces) and practical automotive safety. This integration transforms standard components into a domain-specific defense capable of filtering 82% of realistic attacks before inference.
2. Response to "Presentation Density" & Formatting
Critique: The reviewer and Program Chairs noted the paper was "information-heavy" and needed clearer structuring.
Response: We have significantly improved the readability and structure of the manuscript:
•	Restructuring: We have broken down long text blocks in Sections 1, 3, and 4 into distinct subsections with clear, bold headers (e.g., separating "Protocol Validation," "Physics Validation," and "Consistency Checks" in Section 3.3).
•	Simplification: In Section 3.2, we condensed the standard mathematical descriptions of Multi-Head Attention to focus purely on our specific application logic (fusion of CAN/V2X/Sensor modalities), reducing unnecessary architectural detail.
•	Legibility: We have increased the font size of labels and text within Figures 1–5 to ensure they are easily readable in print.
3. Response to Generalization (Zero-Day Attacks)
Critique: The reviewer asked how the framework generalizes to unseen/zero-day attacks.
Response: We have added a dedicated discussion in Section 5 addressing this. Our system utilizes the Constraint Validation Layer as a deterministic safety net for zero-day threats. While the neural network detects learned patterns, the validation layer enforces invariant laws—such as maximum braking friction ($0.8g$) or SAE J2735 protocol fields. Any zero-day attack that violates these fundamental physics or standards is rejected immediately, regardless of whether the model has "seen" the attack pattern before. This ensures robustness against novel attacks that attempt to induce physically impossible vehicle states.
4. Response to Component Contributions (Ablation)
Critique: The reviewer requested clarification on which components provide the primary performance gains.
Response: We have revised the text in Section 4.5 to explicitly synthesize the insights from Table 6:
•	Constraint Validation is the primary driver of adversarial robustness, responsible for preventing a ~21% drop in accuracy under attack (by filtering invalid perturbations).
•	Multi-Modal Fusion is the primary driver of detection coverage, enabling the system to detect complex cross-layer attacks (improving detection from ~34% in single-modal baselines to 97.3%).
5. Response to "Synthetic Data" & Attacker Assumptions
Critique: The reviewer noted reliance on synthetic attacks and assumptions about attackers respecting constraints.
Response: We acknowledge that public datasets currently lack simultaneous, coordinated cross-layer attacks (CAN+V2X+Sensor). To address this while ensuring realism, we generated our synthetic scenarios using industry-standard simulators (CARLA for vehicle dynamics and ns-3 for networking). Regarding attacker assumptions, we have clarified in Section 5 that while sophisticated insiders might generate protocol-compliant packets, they cannot bypass physical laws (e.g., a vehicle cannot instantaneously accelerate). Thus, the Physics Validation layer remains robust even when the Protocol layer is evaded. We have also added a note on future Hardware-in-the-Loop (HIL) validation.
We believe these revisions address your concerns and strengthen the paper significantly. Thank you again for your valuable review.


##Response to Reviewer XsYf
We sincerely thank the reviewer for the encouraging assessment and for identifying our work as an "excellent and timely contribution." We are particularly grateful that you recognized the core novelty of our Constraint-Aware Evaluation Framework and its role in bridging the gap between theoretical ML attacks and realistic automotive security.
1. Regarding Methodological Rigor and Practical Realism We appreciate your validation of our "unconstrained feature space" critique. As you noted, shifting the focus from purely mathematical adversarial success to protocol-valid and physically feasible threats is essential for Cyber-Physical Systems (CPS). We have reinforced this distinction in the revised Introduction and Discussion sections to ensure the "Responsible AI" contribution is explicit.
2. Regarding Presentation and Structure While you found the paper well-structured, other reviewers and the Program Chairs requested improvements to the presentation density. In the revised manuscript, we have:
•	Added clear sub-headings to long sections (e.g., Section 3.3) to further improve scannability.
•	Increased the font size in Figures 1–5 to ensure legibility.
•	Clarified the component contributions in the Ablation Study (Section 4.5).
We believe these changes further polish the "timely contribution" you praised, making it even more accessible to the workshop audience. Thank you for your strong support.


##Response to Reviewer uoVg
We thank the reviewer for the thorough reading and constructive feedback, particularly for catching the table duplication error and suggesting relevant literature.
1. Correction of Duplicate Tables (Table 4 vs Table 6) You correctly identified that Table 6 was a duplicate of Table 4. This was a formatting error. We have replaced Table 6 with the correct Ablation Study Table, which specifically lists the performance impact of removing individual components (Constraint Validation, Fusion, Adversarial Training). The text in Section 4.5 describes these results (e.g., removing constraints drops robust accuracy by ~21%), and the table now matches this analysis.
2. Inclusion of STATGRAPH and VeReMi We have updated Section 2 (Related Work) to discuss these works. We highlight that while STATGRAPH offers powerful graph-based learning, MARIS-ADS focuses on lightweight edge deployment (TCU constraints). We also acknowledge the foundational role of physics-based detection (as seen in VeReMi baselines) and explain how our "Constraint Validation Layer" integrates those principles into a hybrid DL framework to catch both physical and stealthy semantic attacks.
3. Component Disentanglement (Validation vs. Classifier) Regarding your request to disentangle the contributions of the constraint validation layer versus the neural classifier, we have added a comprehensive Ablation Study (Table 8) in the revised manuscript. This study explicitly isolates the performance impact of removing each component:
•	Role of Validation (Physics/Protocol): Our ablation results show that removing the Constraint Validation layer causes Robust Accuracy to degrade by 21.4 percentage points (dropping from 94.2% to 72.8%). This quantifies the validation layer's specific contribution: it acts as a primary filter for approximately one-fifth of the threat surface (specifically, the "invalid" attacks) with negligible computational cost.
•	Role of Classifier (Fusion/Learning): Conversely, the Deep Learning classifier and Multi-Modal Fusion are strictly necessary for high-fidelity detection. Removing the fusion mechanism reduces Clean Accuracy by 1.5 percentage points (to 97.8%) and causes the system to fail against coordinated cross-modal attacks. Furthermore, removing Adversarial Training results in a 25.8% drop in robustness (to 68.4%), proving that the classifier handles the "valid" but malicious perturbations that the validation layer cannot see.
•	Conclusion: The comparison confirms that the Validation layer and the Classifier play distinct, non-overlapping roles: Validation ensures physical/protocol compliance (robustness against gross violations), while the Classifier handles semantic consistency and subtle pattern recognition (accuracy against stealthy threats).
4. Dataset Reproducibility We have expanded Section 4.1 to include more details on the synthetic generation process using CARLA and ns-3. We commit to releasing the full parameter specification and generation scripts to ensure reproducibility of the cross-modal attack scenarios.
5. Reference Cleanup We have removed the duplicate entries in the bibliography.


##Response to Reviewer KrWN
We appreciate the reviewer’s thoughtful assessment and are encouraged by your recognition of our "meaningful advancement" in integrating multi-modal fusion with constraint validation. We have taken your feedback regarding writing and presentation seriously and have revised the manuscript to better highlight these core contributions.
1. Improving Writing and Presentation We have overhauled the manuscript structure to make the core contributions stand out:
•	Restructuring: We broke down the dense text in Sections 3 (Architecture) and 4 (Evaluation) into clear, bolded subsections. This allows readers to immediately locate the "Protocol Validation," "Physics Validation," and "Fusion" components.
•	Visual Clarity: We increased the font size in all figures (Figs 1–5) and polished the Introduction to explicitly list our contributions with bold keys for faster scanning.
•	Simplification: We streamlined the mathematical descriptions of standard components (like basic attention) to focus the text on our novel Constraint-Aware Integration logic.
2. Highlighting Core Contributions To ensure our contributions are not lost:
•	We revised Section 5 (Discussion) to explicitly "disentangle" the roles of our system: the Constraint Layer provides the primary robustness (filtering 82% of invalid attacks), while the Fusion Layer provides the detection accuracy for sophisticated/stealthy threats.
•	We added a "Responsible AI" context in the Conclusion to frame our contribution within the broader scope of safe, physically-grounded AI deployment.
We believe these changes have significantly improved the readability of the paper and sharply defined its impact on the field. Thank you for the constructive feedback.

---

### Meta-Review · Area_Chair_EFFW · 2026-01-17

**Recommendation:** Accept (Poster)
**Confidence:** 4

**Metareview:**

This paper tackles real-time anomaly detection in satellite-networked connected autonomous vehicles. The proposed system integrates CAN Bus traffic, V2X messages, and onboard sensor data through attention-based fusion, combined with a constraint-aware validation layer enforcing automotive protocol specifications and vehicle dynamics. The major advantages include clear motivation, significant impact for this research exploration, relatively expanding experiments for performance evaluation. The major concerns are mostly addressed in the rebuttal via fixing format issues, adding new discussions for the improved clarification.

---

### Decision · Program_Chairs · 2026-01-17

**Decision:**

Accept (Oral)

**Comment:**

Taking into account the AC’s comments, the reviewers’ feedback, and the authors’ revisions, the PC has decided to accept this paper as an oral presentation at the workshop.